# Experimental Study for the Sorption and Diffusion of Supercritical Carbon Dioxide into Polyetherimide

**DOI:** 10.3390/molecules29174233

**Published:** 2024-09-06

**Authors:** Wei-Heng Huang, Pei-Hua Chen, Chin-Wen Chen, Chie-Shaan Su, Muoi Tang, Jung-Chin Tsai, Yan-Ping Chen, Feng-Huei Lin

**Affiliations:** 1Department of Chemical and Materials Engineering, Chinese Culture University, Taipei 111396, Taiwan; a6580800@ulive.pccu.edu.tw (W.-H.H.); muoitang@ulive.pccu.edu.tw (M.T.); 2Department of Biomedical Engineering, National Taiwan University, Taipei 106319, Taiwan; 16185@s.tmu.edu.tw (P.-H.C.); double@ntu.edu.tw (F.-H.L.); 3Department of Orthopedics, Shuang Ho Hospital, Taipei Medical University, New Taipei City 235041, Taiwan; 4Department of Molecular Science and Engineering, National Taipei University of Technology, Taipei 106344, Taiwan; 5Department of Chemical Engineering and Biotechnology, National Taipei University of Technology, Taipei 106344, Taiwan; cssu@ntut.edu.tw; 6Department of Chemical Engineering, Ming Chi University of Technology, New Taipei City 243303, Taiwan; jctsai@mcut.edu.tw; 7Department of Chemical Engineering, National Taiwan University, Taipei 106319, Taiwan; ypchen@ntu.edu.tw

**Keywords:** supercritical carbon dioxide, polyetherimide, sorption, Fickian diffusion model, diffusivity

## Abstract

Supercritical carbon dioxide (SCCO_2_) is a non-toxic and environmentally friendly fluid and has been used in polymerization reactions, processing, foaming, and plasticizing of polymers. Exploring the behavior and data of SCCO_2_ sorption and dissolution in polymers provides essential information for polymer applications. This study investigated the sorption and diffusion of SCCO_2_ into polyetherimide (PEI). The sorption and desorption processes of SCCO_2_ in PEI samples were measured in the temperature range from 40 to 60 °C, the pressure range from 20 to 40 MPa, and the sorption time from 0.25 to 52 h. This study used the ex situ gravimetric method under different operating conditions and applied the Fickian diffusion model to determine the mass diffusivity of SCCO_2_ during sorption and desorption processes into and out of PEI. The equilibrium mass gain fraction of SCCO_2_ into PEI was reported from 9.0 wt% (at 60 °C and 20 MPa) to 12.8 wt% (at 40 °C and 40 MPa). The sorption amount increased with the increasing SCCO_2_ pressure and decreased with the increasing SCCO_2_ temperature. This study showed the crossover phenomenon of equilibrium mass gain fraction isotherms with respect to SCCO_2_ density. Changes in the sorption mechanism in PEI were observed when the SCCO_2_ density was at approximately 840 kg/m^3^. This study qualitatively performed FTIR analysis during the SCCO_2_ desorption process. A CO_2_ antisymmetric stretching mode was observed near a wavenumber of 2340 cm^−1^. A comparison of loss modulus measurements of pure and SCCO_2_-treated PEI specimens showed the shifting of loss maxima. This result showed that the plasticization of PEI was achieved through the sorption process of SCCO_2_.

## 1. Introduction

Studying the sorption behavior of gas or penetrant in synthetic polymers is of great importance for fundamental research and technological applications and has been discussed in recent reviews [1,2,3,4]. Many important developments exist in applying polymer materials with low- or high-pressure carbon dioxide (CO_2_), as described below. Polymer-containing materials have been used in zero-emission pathways for CO_2_ capture or CO_2_ transport chain, where gas sorption, selectivity, and polymer modification are interesting topics [5,6]. Regarding material processing, the sorption of supercritical CO_2_ (SCCO_2_) into polymers is of great importance for the surface modification or foaming of substrates. SCCO_2_ has widely been used in the above processing of polymer materials due to its good diffusion coefficient, low viscosity, and safety considerations [7,8]. Modifiers are injected into the polymer substrate through sorption of SCCO_2_, which is the first step to be considered when controlling polymer properties such as surface grafting. Foaming and surface modification of polymers using SCCO_2_ as the blowing agent or as a green solvent for reaction medium to produce medical or biocompatible value-added products have recently been reviewed [9,10], where SCCO_2_ solubility data are essentially needed. Surface graft polymerization is an example of minimizing protein fouling in protein recovery application [11] or enhancing the hydrophilicity of versatile commodity polymers [12]. Processing polymeric materials requires information on the phase equilibrium and transport properties of SCCO_2_ in the polymer matrix. The polymer modification strategy as an oxygenator in extracorporeal respiratory circulation or grafting hydrophilic 2-hydroxyethyl methacrylate (HEMA) monomer onto polyacrylonitrile (PAN) polymer substrate to obtain ultrafiltration biomedical materials requires information on the sorption and diffusion between SCCO_2_ and polymers [10,13,14].

The sorption and transport of SCCO_2_ in polymers was recently studied theoretically by Ricci et al. [15]. Experimental studies for the sorption amounts and diffusion coefficients of SCCO_2_ in various polymer substrates have been reported in the literature [16,17,18,19,20,21,22,23,24,25]. The recent literature shows that the availability of thermodynamic and transport properties (e.g., solubility and diffusivity) of SCCO_2_–polymer mixtures is limited [15], and more experimental data on various polymer systems are still needed. The gravimetric technique is the most common method used to measure the amount and rate of SCCO_2_ sorption into polymers. The gravimetric method records the rate of weight change in a polymer sample after soaking in SCCO_2_ at a given temperature, pressure, and sorption time. The Fickian mass transfer model then analyzes the recorded data to evaluate the diffusivity of SCCO_2_ into or out from the polymer or copolymer membrane samples [24,25]. The results of these fundamental studies supply essential information for SCCO_2_-assisted polymer processing, as shown in a previous review article [26]. Fundamental data on the sorption and desorption diffusivities of SCCO_2_ in many unstudied polymers, copolymers, or polymer blend systems are still needed. These data can enable new applications of polymer modification to produce value-added products such as drug delivery foams and films, biomedical devices, and biodegradable materials in tissue engineering [4].

In this study, we report the experimental measurement results for the sorption and diffusion of SCCO_2_ into polyetherimide (PEI). PEI is an amorphous thermoplastic polymer with chemical stability and ductile properties for various applications. It has been illustrated that PEI is a candidate for biomedical usage, such as intraocular lenses, biosensors, or neuroprostheses [27]. The PEI backbone can be modified through wet chemistry to prepare membranes for artificial organs. Feng et al. [28] have presented using SCCO_2_/ethanol co-foaming technology to fabricate PEI bead foams with three-dimensional geometry for special engineering plastic materials in high-tech industries. The production of PEI nanofoams has been presented by Aher et al. [29] using SCCO_2_ as the blowing agent. They studied the sorption of SCCO_2_ at 20 MPa into the commercially available PEI sheets. The diffusion coefficients at 0 °C and room temperature (23 °C) were experimentally determined. The equilibrium concentration of SCCO_2_ absorbed in PEI has also been studied by Zhou et al. [30] at pressures from 6 to 10 MPa. They conducted the SCCO_2_ saturation study in PEI before fabricating the PEI nanofoams with high strength, toughness, and good thermal resistivity. A process map for the foaming temperature and absorbed SCCO_2_ concentration was developed.

This study aimed to investigate the novel sorption and diffusion of SCCO_2_ in PEI using the ex situ gravimetric method within a pressure and temperature range (pressure studied at 20, 30, and 40 MPa; temperature studied at 40, 50, and 60 °C). The mass diffusivities were evaluated based on the Fickian diffusion model using experimentally measured data under various experimental conditions. The effects of temperature and pressure on the sorption mass gain fractions and sorption diffusivities are discussed. The equilibrium sorption mass gain fractions depended on the density of SCCO_2_, where the crossover phenomenon of the absorption isotherms was determined and explained. The desorption behavior of SCCO_2_ from PEI was analyzed through Fourier-transform infrared (FTIR) spectroscopy, demonstrating the antisymmetric stretching mode of CO_2_ trapped in PEI at different desorption times. The plasticization effect of PEI by absorbed SCCO_2_ was examined using loss modulus measurements, where the measured results of loss maxima were illustrated. These basic data provide helpful information for future applications of PEI, such as being a prime candidate for medical tools [27,31] with good biocompatibility and becoming an integral component of interior panels, electrical enclosures, and aerospace industry [9] due to its flame retardancy and weight reduction potential.

## 2. Results

### 2.1. Determination of the Sorption and Desorption Diffusivities of SCCO_2_ in PEI

For the sorption experiments in this study, we used the ex situ gravimetric method, and the detailed description of the experimental apparatus and procedures is shown in Section 3.2. In the experiments, we measured the desorption mass gain fraction (*M_d_*), determined the sorption mass gain fraction (*M_s_*) and the equilibrium sorption mass gain fraction (*M_∞_*) at specific operation conditions. Figure 1 presents the plot of the desorption mass gain fraction *M_d_* against the square root of desorption time *t_d_*, where the sorption experiments were performed at 60 °C, 30 MPa, and various sorption time parameters *t_s_*. The different desorption lines shown in Figure 1 were obtained from different PEI sample sheets during the experiments. Detailed experimental procedures are described in Section 3.2, Apparatus and procedures. It is stated in Section 3.3, Data analysis method, that the linear relations between *M_d_* with the square root of the desorption time (*t_d_*)^1/2^, as represented by Equation (5), demonstrated that the Fickian diffusion model [32] was suitable to fit the experimental data at a short desorption interval. The mass gain fraction of SCCO_2_ (*M_s_*) absorbed into the PEI sample at the specific sorption time *t_s_* parameter was determined as the intercept by extrapolating each desorption line to zero desorption time. This extrapolation method is described in Section 3.2, Apparatus and procedures.

By repeating the desorption experiments using the ex situ gravimetric method under different operating temperatures, pressures, and various sorption time parameters, the sorption mass gain fractions (*M_s_*) under specific sorption conditions were obtained. Figure 2 shows the results at 60 °C and three sorption isobars at 20, 30, and 40 MPa, respectively. For PEI samples with a thickness of 0.6 mm, the SCCO_2_ sorption amount leveled off after reaching equilibrium around 50 h under all experimental conditions. The leveled-off point yielded experimental data for the equilibrium mass gain fraction, *M_∞_*. The regression results of the sorption curves for each isobar were satisfactory and are displayed in Figure 2.

The *M_∞_* values were determined by plateauing the isobaric *M_s_* data, and these values support the necessary information for further polymer processing, such as foaming or grafting. This study also obtained similar experimental results for three isobars at 40 and 50 °C, respectively. All these experimental data were used to evaluate the sorption diffusivity *D_s_* and desorption diffusivity *D_d_* for long sorption time and short desorption time, respectively. A linear plot method was employed to determine the diffusivity data described in Equations (4) and (5) in Section 3.3. Table 1 lists the diffusivity results for SCCO_2_ sorption and desorption in PEI for three isobars at 40, 50, and 60 °C, respectively. The density data listed in Table 1 for pure CO_2_ were retrieved from the NIST database [33]. It can be observed from Table 1 that at the lowest temperature and highest pressure of 40 °C and 40 MPa, the highest sorption mass gain fraction of SCCO_2_ in PEI is 12.8 wt%. The *M_∞_* values measured in this study are consistent with values reported in the literature [29], which reported that *M_∞_* for PEI was approximately 10 wt% at 20 MPa and 45 °C.

Figure 3 shows a graphical representation of the equilibrium sorption mass gain fraction (*M_∞_*) for three isotherms at different pressures, including all experimental data in this study. The equilibrium sorption mass gain fraction at a constant temperature increased with the increasing pressure. Taking the 60 °C isotherm as an example, *M_∞_* of 7.9 wt% was obtained at a low pressure of 13.5 MPa, and *M_∞_* of 14.1 wt% was obtained at a high pressure of 58.3 MPa. Figure 3 also indicates that *M_∞_* decreased with increasing temperature at constant pressure. The literature mentions that the spectroscopic results showed that CO_2_ has specific interactions with various polymers. This interaction has exothermic properties, resulting in reduced solubility of CO_2_ in the polymer at higher temperatures under isobaric conditions [20,34,35,36]. The same trend was found in the literature on SCCO_2_ sorption in polycarbonate (PC) and polysulfone (PSF) [24,25], as well as poly(vinyl chloride) (PVC) [37]. Compared with the experimental results in the literature, the *M_∞_* values of polycarbonate (PC) polymer were relatively higher than those of polysulfone (PSF) [24,25] and PEI. Under the same temperature and pressure conditions, the *M_∞_* values of PEI measured in this study are close to those of PSF. The interactions between CO_2_ and functional groups in various polymer matrices may be responsible for these findings. The interaction between CO_2_ and the carbonyl groups of PC increased the amount of CO_2_ absorbed in the polymer, as described in the literature, where this interaction has been studied using experimental or theoretical methods [38,39,40,41,42,43].

### 2.2. The Sorption Mechanism of SCCO_2_ in PEI

Furthermore, Figure 4 shows a plot of *M_∞_* versus SCCO_2_ density for the three isotherms in this study, where concave upward curves were observed. The continuous curves in Figure 4 were obtained by optimal regression of the experimental results. The crossover phenomena is observed in Figure 4. According to the previous literature studies [44,45], the change in the slope of the isotherm shown in Figure 4 may indicate that the PEI polymer has been plasticized. Figure 4 also shows the crossover point with a SCCO_2_ density of approximately 840 kg/m^3^. At lower densities below the crossover point, the solubility of SCCO_2_ in PEI decreased with increasing temperature. SCCO_2_ solubility increased with temperature when the density was above the crossover point. When the SCCO_2_ density exceeds 840 kg/m^3^, the 40 and 50 °C isotherms are close to each other, but the 50 °C data are still slightly higher than the 40 °C data. The same concave upward curves of SCCO_2_ solubility isotherms were also found in previous studies for polymers of PC and poly (ethylene terephthalate) (PET) [24,44].

Crossover phenomena show the transfer of sorption mechanisms in polymers of different density states. Gas sorption in polymers has been discussed and reviewed through various physical and mathematical models [46,47,48,49]. Experimentally measured gas sorption and desorption data in polymers can be interpreted based on these theoretical considerations. In the lower-density region of SCCO_2_, the CO_2_ molecular sorption model was dominated by the solubility of CO_2_ in the glassy state of the polymer. The density or solvent power was higher for SCCO_2_ at lower temperatures, leading to a larger equilibrium sorption mass gain fraction (*M_∞_*). When the SCCO_2_ density exceeded the crossover point or the penetration concentration [44] where the glass transition occurred, the mobility of the polymer chains increased to a higher degree due to the greater sorption of SCCO_2_. With increasing temperature, more plasticization effects existed in higher density regions, as shown in Figure 4. Similar crossover behavior was also expressed in the literature on SCCO_2_ sorption in PC and PET [24,44]. In the lower-density region, it is assumed that dual-mode absorption was appropriate for the mass transfer mechanism, where SCCO_2_ was absorbed up to the second layer of the polymer substrate. At higher densities beyond the crossover point, the polymer may transit from a glassy to a rubbery state, producing a Fickian diffusion pattern. However, the crossover density value depends on the glass transition temperature (*T_g_*) of the various polymers. PET had a crossover density of approximately 400 kg/m^3^ and a *T_g_* of approximately 75 °C [44,50], while PC had a crossover density of approximately 680 kg/m^3^ and a *T_g_* of approximately 150 °C [21,51,52]. The measured *T_g_* of the PEI sample used in this study was approximately 217 °C, consistent with the literature data [53]. Due to the higher *T_g_*, PEI reasonably corresponded to a higher crossover density of 840 kg/m^3^.

### 2.3. Comparison of SCCO_2_ Diffusivities in Various Polymers

According to Equation (5) presented in Section 3.3, the desorption diffusivity *D_d_* was evaluated from the linear slope values from the plots of (*M_d_/M_s_*) against (*t_d_*)^1/2^. The plot of *D_d_* against *M_∞_* is shown in Figure 5 for three temperatures with the experimental data obtained in this study. The continuous curve in Figure 5 was obtained by optimal regression of the experimental results. It is observed from Figure 5 that the desorption diffusivities increased significantly with increasing equilibrium SCCO_2_ mass gain fraction in the PEI polymer substrate. Under the conditions of 40 °C and 40 MPa, the *D_d_* of SCCO_2_ in the PEI matrix was 1.47 × 10^−11^ m^2^/s, in which the sorption mass gain fraction of SCCO_2_ was 12.8 wt%. The *D_d_* measured in this study at 60 °C and 13.5 MPa was 0.23 × 10^−11^ m^2^/s, in which the sorption mass gain fraction of SCCO_2_ was 7.9 wt%. This trend is in agreement with the results of the literature on SCCO_2_ desorption in PVC, PC, and PSF polymers [24,25,37].

As given in Table 1, *D_s_* values increased with temperature but had relatively little dependence on pressure. The largest *D_s_* value was 0.30 × 10^−11^ m^2^/s at 60 °C and 20 MPa. Sorption at higher temperatures accelerated CO_2_ molecules to fill into the sites of the polymer substrate with higher kinetic energy and, therefore, increased *D_s_*. Due to higher polymer chain mobility at higher temperatures, the driving force of mass transfer would rise, and the drag force for the motion of CO_2_ molecules in the polymer would decrease.

PEI exhibited the lowest sorption and desorption diffusivities compared to PC and PSF, which may be due to the physical properties of these polymers. The glass transition temperature of PEI is up to 217 °C [53], which is higher than 150 °C for PC [21,51,52] and 185 to 187 °C for PSF [54,55]. Studies have also found that the yield strength of PEI was 100 to 110 MPa (room temperature) [56,57]. The yield strengths of PC and PSF were approximately 65 MPa and 75 MPa, respectively [58,59]. It can also be observed from Table 1 that the *D_d_* values were greater than *D_s_*, and other polymers of PSF and PC also showed a similar trend [24,25]. During the sorption process, dissolved CO_2_ must overcome the interaction forces between polymer chains. The desorption process was carried out under atmospheric pressure; there was a significant pressure drop, and the polymer matrix has also been plasticized during the sorption process. This is the reason why *D_d_* is larger than *D_s_* for PC, PSF, and PEI polymers. However, Muth et al. [37] reported that *D_s_* values were larger than *D_d_* for PVC polymer. This might be due to the fact that PVC has relatively smaller yield strength (about 45 MPa) and lower *T_g_* (about 82 °C) [60], which was favorable for gas sorption.

### 2.4. Plasticization Effect for the Sorption of SCCO_2_ in PEI

Plasticization refers to changes in a given polymer’s thermal or mechanical properties, including a decrease in its stiffness and a decrease in its glass transition temperature. Polymer processing using SCCO_2_ allows for control of polymer properties such as viscosity and plasticity. In the above discussion of the sorption and desorption diffusivities of CO_2_ in PC, PSF, and PEI, the plasticization effect of absorbed CO_2_ explains the differences in diffusivity values. During the sorption process, the polymer matrix initially existed in a state of entangled bonds. CO_2_ required a stronger driving force for mass transfer to overcome the greater resistance due to the lower mobility of the polymer chains. During desorption, the polymer matrix was swollen and plasticized by CO_2_ to have higher chain mobility. CO_2_ molecules experienced less resistance, thereby increasing the desorption rate of CO_2_ from the polymer matrix. The existence of CO_2_ in PEI was qualitatively investigated in this study by examining the characteristic FTIR spectra. Figure 6 compares the FTIR spectra of untreated PEI and PEI desorbed for 24 h after SCCO_2_ sorption treatment for 12 h at 20 MPa and 40 °C. The trapped CO_2_ within the PEI specimen can be observed from the spectra with the CO_2_ bending mode (ν_2_) near 660 cm^−1^ and the antisymmetric stretching mode (*ν*_3_) near 2340 cm^−1^.

Figure 7 shows the antisymmetric stretching pattern (*ν*_3_) around 2340 cm^−1^ for trapped CO_2_ in PEI specimens at different desorption times. Spectra (a) and (i) represent FTIR results for pure gaseous CO_2_ and PEI, respectively. As shown in spectra (b) to (d), the bands appear to have very broad widths over the desorption interval of 120 s to 4 h. As the desorption time increased, the decrease in transmission bandwidth indicated the desorption of CO_2_ from PEI. The IR transmittance shows that CO_2_ still existed in the PEI substrate 72 h after being discharged from the high-pressure cell, and the peak in the spectrum (g) still appeared near the wavenumber at 2340 cm^−1^. The phenomena in Figure 7 qualitatively shows that CO_2_ remained in PEI after various desorption times.

The SEM images of the untreated PEI and the SCCO_2_-treated PEI (under the process condition of 40 °C, 20 MPa, and sorption time for 12 h) are presented in Figure 8. Compared to the untreated PEI in Figure 8a, microstructure change and surface deformation were observed for SCCO_2_-treated PEI in Figure 8b. This morphology change also qualitatively implies the plasticization effect during the sorption and desorption of SCCO_2_ in PEI.

Figure 9 shows loss modulus curves of (a) untreated PEI, (b) PEI after sorption in SCCO_2_ at 20 MPa and 40 °C for 12 h, and (c) PEI under previous SCCO_2_-treated conditions that had been depressurized in the atmosphere for more than one month. The maximum loss modulus of untreated PEI was 233 °C, while that of treated PEI was 227 °C. This shift in loss maxima was attributed to the plasticization of the PEI substrate with the increase in polymer chain mobility. This result also corresponds to the fact that *D_d_* was greater than *D_s_* for SCCO_2_ in PEI. Moreover, when the SCCO_2_-treated PEI was depressurized for over one month, the maximum loss modulus returned to 232 °C, nearly the same as untreated PEI. It indicated that SCCO_2_ left no residual in PEI after a long time of depressurization and would have little effect on PEI’s properties.

## 3. Materials and Methods

### 3.1. Material

Polyetherimide (PEI, CAS registry number 61128-46-9, molecular formula (C_37_H_24_N_2_O_6_)_n_, melt index 9 g/10 min (at 337 °C, 6.6 kg), density 1.27 g/cm^3^, glass transition temperature 217 °C) was purchased from Sigma-Aldrich, UNI-ONWARD Corp., New Taipei City, Taiwan. The structure of PEI is shown in Figure 10. PEI was hot pressed at 240 °C and then cut into dimensions of 40 mm × 10 mm. The membrane sample with a thickness of 0.6 mm for experimental sorption measurements and the membrane samples with a thickness of 0.2 mm were used for FTIR and loss modulus experiments. CO_2_ was purchased from San-Fu Chemical Company, Taiwan, with a certified purity greater than 99.8%. All chemicals were used as received.

### 3.2. Apparatus and Procedures

The experimental equipment is shown in Figure 11. The CO_2_ was stored in the cylinder and compressed to the operating pressure by a syringe pump (ISCO, model 100DX, ISCO, Lincoln, NE, USA). The PEI specimen was weighed using a digital balance (Mettler AE200, Greifensee, Switzerland, sensitivity 0.1 mg) prior to putting it into a stainless steel high-pressure cell with an inner diameter of 5/8 inch and capacity of 10 cm^3^. The high-pressure cell was maintained at a desired temperature using a constant temperature bath (ISCO, SFX2-10).

At the beginning of the ex situ gravimetric experiment, the original PEI specimen was weighed using a digital balance to record its initial weight, *M*_0_. The high-pressure cell was purged with pure CO_2_ to remove any air inside. Then, with the outlet valve closed, pressurized CO_2_ from the ISCO injection pump was charged into the high-pressure equilibrium cell. The equilibrium cell reached the preset equilibrium pressure within 10 s.

The pressure and temperature in the cell were dynamically controlled throughout the sorption experiments. After a certain period of sorption time, the cell was depressurized, and the specimen was rapidly taken onto the digital balance at room temperature under atmospheric pressure. The originally dissolved CO_2_ in the PEI specimen was desorbed as pressure was released. The PEI specimen’s weight was recorded by the digital balance as a function of time. Normally, it takes 30 s during the depressurization process before the digital balance records the first data. At desorption time *t_d_*, the weight of the PEI membrane was *M*. The mass gain fraction was calculated by
(1)$$\mathrm{Mass}\;\mathrm{gain}\;\mathrm{fraction}\;(\mathrm{wt}\%)=\frac{M-M_{0}}{M_{0}}\times 100\%$$

During the desorption process, the weight of the PEI sample was measured at every 10-s interval for at least 120 s. These measurement procedures were similar to those in the previous literature, where the ex situ gravimetric method was applied [24,25]. A PEI sample sheet was used only once at a specific sorption temperature, pressure, and sorption time. A number of PEI sample sheets were prepared and used in this study. The weight of the PEI sample decreased with the prolongation of desorption time, and the mass gain fraction curve showed a linear decreasing trend with the square root of desorption time within the first 100 s. A schematic sorption and desorption diagram is shown in Figure 12.

It is presented in Figure 12 that at a sorption or soaking time *t_s_*, the mass gain fraction of the PEI specimen owing to the sorption of CO_2_ was *M_s_*. In the desorption process, the mass gain fraction decreased with the extension of desorption time and became *M_d_* at a certain desorption time *t_d_*. For an initial desorption time of about 100 s, the reduction in mass gain fraction is almost a linear function of the square root of the desorption time *t_d_*, as shown by the graph inserted in Figure 12. The *M_s_* value was determined by extrapolating the linear portion of the initial desorption curve to zero desorption time, as shown by the point *M_s_* (with the symbol Δ) in the graph inserted in Figure 12. At a long enough sorption time, the mass gain fraction reached its saturation value *M_∞_* corresponded to the equilibrium mass gain fraction of SCCO_2_ at the given experimental temperature, pressure, and sorption time. These data were then utilized to calculate sorption and desorption diffusivities of SCCO_2_ in polymer. A FTIR spectrometer (Digilab FTS-3000, Burladingen, Germany) at room temperature and atmospheric pressure with a resolution of 2 cm^−1^ was used to analyze the transmittance bands of CO_2_ in the PEI specimen. A loss modulus analysis (TA instruments DMA2980, New Castle, DE, USA) was used to determine the shifting of loss maxima for pure and SCCO_2_-treated PEI specimens.

### 3.3. Data Analysis Method

The analyses of sorption and desorption of SCCO_2_ in polymer have been presented in the previous literature [14,23,24,25], based on the mathematical model investigated by Fick. According to the one-dimension Fickian diffusion model, the governing equation for the mass transfer of SCCO_2_ in a plane sheet polymer is
(2)$$\frac{\partial C}{\partial t}=D\frac{\partial^{2}C}{\partial X^{2}}$$
where *C* is the CO_2_ concentration in PEI, and the CO_2_ diffusivity *D* is assumed to be constant at a specific temperature and pressure. The solution to Equation (2) is based on the assumption of semi-infinite or one-dimensional sheets in most cases. According to the literature, for thin flat membrane geometries, the thickness-to-length ratio should be less than 0.16 [20,21]. The ratio in this study was 0.06, which is suitable for applying the one-dimensional solution of Equation (2) without considering the edge effect. Applying the approach of Laplace transform with proper boundary conditions, the solution of Equation (2) is [32]
(3)$$\frac{M_{s}}{M_{\infty }}=1-\frac{8}{\pi^{2}}\sum_{n=0}^{\infty }\frac{1}{{\left(2n+1\right)}^{2}}exp[\frac{-{\left(2n+1\right)}^{2}\pi^{2}D_{s}t_{s}}{l^{2}}]$$
where *l* is the thickness of the PEI sample membrane, *D_s_* is the sorption diffusivity, *M_s_* and *M_∞_* are the sorption mass gain fraction at sorption time *t_s_* and the equilibrium sorption mass gain fraction, respectively. For a long sorption process, Equation (3) is simplified as
(4)$$\frac{M_{s}}{M_{\infty }}=1-\frac{8}{\pi^{2}}exp[\frac{-\pi^{2}D_{s}t_{s}}{l^{2}}]$$

The sorption diffusivity *D_s_* can thus be evaluated by plotting of ln [1 − (*M_s_*/*M_∞_*)] against *t_s_*/*l*^2^.

Equation (2) can also be used to solve the desorption process for a thin flat film polymer. After simplifying the solution to a short desorption time *t_d_*, the mass gain fraction of desorption *M_d_* is expressed by
(5)$$\frac{M_{d}}{M_{s}}=-\frac{4}{l}\sqrt[{}]{\frac{D_{d}t_{d}}{\pi }}$$
where *D_d_* is the diffusivity for desorption. As mentioned above, the *M_s_* value is determined by extrapolating the linear portion of the short-time desorption curve to zero desorption time. The short time period desorption diffusivity *D_d_* can also be obtained from data plotting the desorption mass gain fraction (*M_d_/M_s_*) versus (*t_d_*)^1/2^.

## 4. Conclusions

In this study, the sorption and diffusion experiments of SCCO_2_ in 0.6 mm thick PEI membrane samples were conducted in the temperature and pressure ranges of 40 to 60 °C and 20 to 40 MPa, respectively. A high-pressure equilibrium cell and the ISCO units were used in this study. The sorption amounts under different operating conditions were determined using an ex situ gravimetric method. The sorption and desorption diffusivities were evaluated using the Fickian diffusion model. The equilibrium sorption mass gain fraction depended on the process conditions and reached the value of 12.8 wt% at 40 °C and 40 MPa. The sorption diffusivity increased with temperature and was slightly pressure-dependent. Due to the plasticization effect, the desorption diffusivity was greater than the sorption diffusivity in PEI. The plot of sorption mass gain fractions at isotherms against SCCO_2_ density showed a crossover point at the SCCO_2_ density of approximately 840 kg/m^3^. The result indicated changes in the mass transfer mechanism at the crossover point. This study compared the crossover densities of different polymers and derived the dependence of the glass transition temperatures and yield strengths of different polymer substrates. The desorption of CO_2_ from PEI was also qualitatively studied using FTIR spectroscopy, and a decrease in absorbed CO_2_ in the PEI polymer was observed with increasing desorption time. This study examined the shift in the loss modulus maxima and the SEM images of SCCO_2_-treated PEI, which was indicative of plasticization on PEI induced by SCCO_2_ sorption.

## Figures and Tables

**Figure 1 molecules-29-04233-f001:**
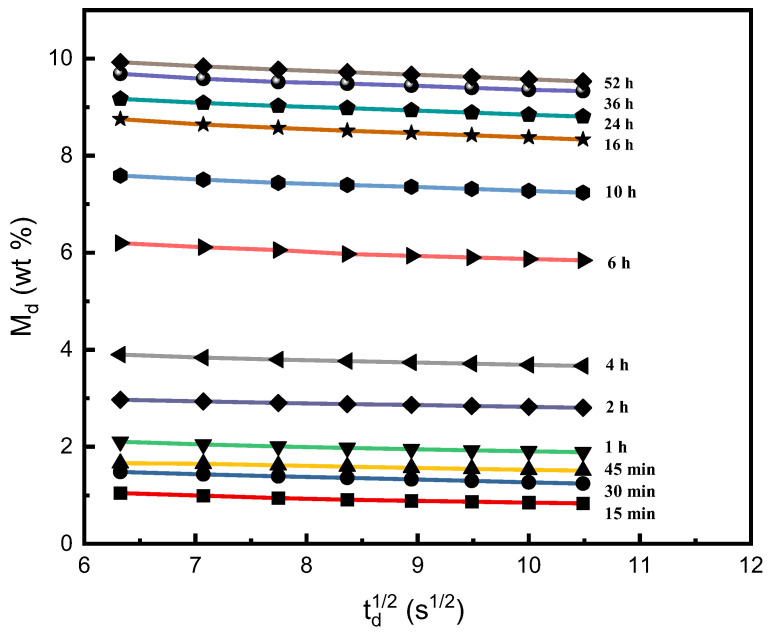
Plot of the desorption mass gain fraction (*M_d_*) against the square root of the desorption time (*t_d_*)^1/2^ for PEI, where the sorption experiments were performed at 60 °C, 30 MPa, and various sorption time parameters *t_s_* from 15 min to 52 h.

**Figure 2 molecules-29-04233-f002:**
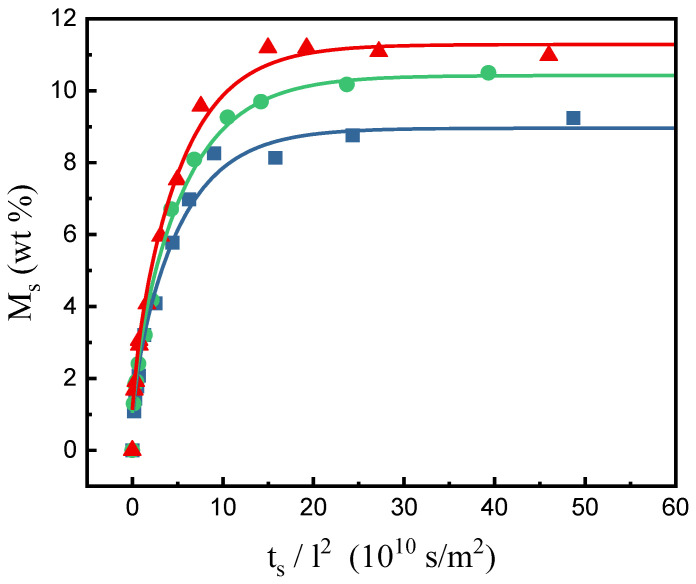
The mass gain fractions of PEI at 60 °C and various pressures: ■, 20 MPa; ●, 30 MPa; ▲, 40 MPa. The curves were obtained by regression on the experimental data.

**Figure 3 molecules-29-04233-f003:**
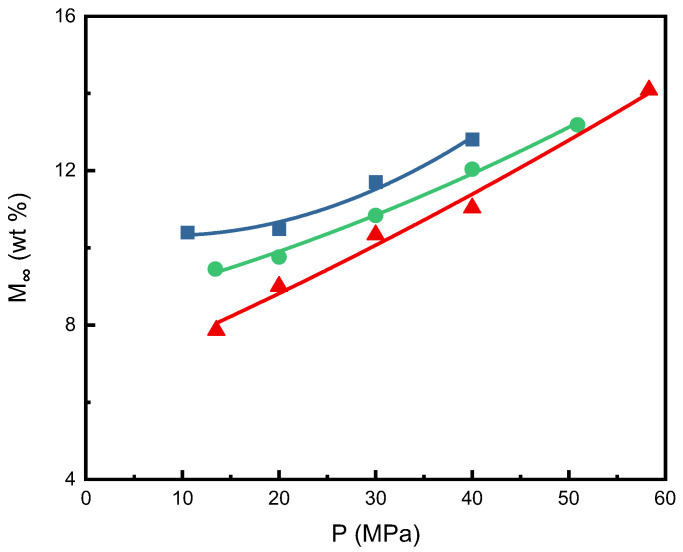
Plot of the equilibrium sorption mass gain fraction of SCCO_2_ (*M_∞_*) against pressure at various temperatures: ■, 40 °C; ●, 50 °C; ▲, 60 °C.

**Figure 4 molecules-29-04233-f004:**
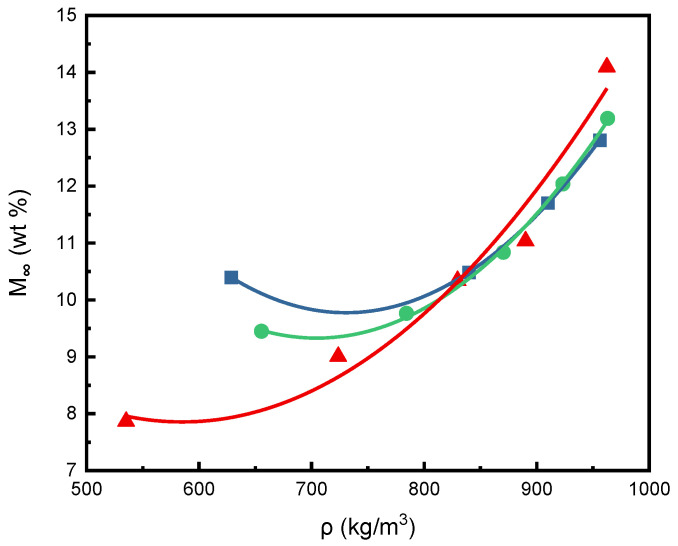
Plot of the equilibrium sorption mass gain fraction of SCCO_2_ (*M_∞_*) against SCCO_2_ density at various temperatures: ■, 40 °C; ●, 50 °C; ▲, 60 °C.

**Figure 5 molecules-29-04233-f005:**
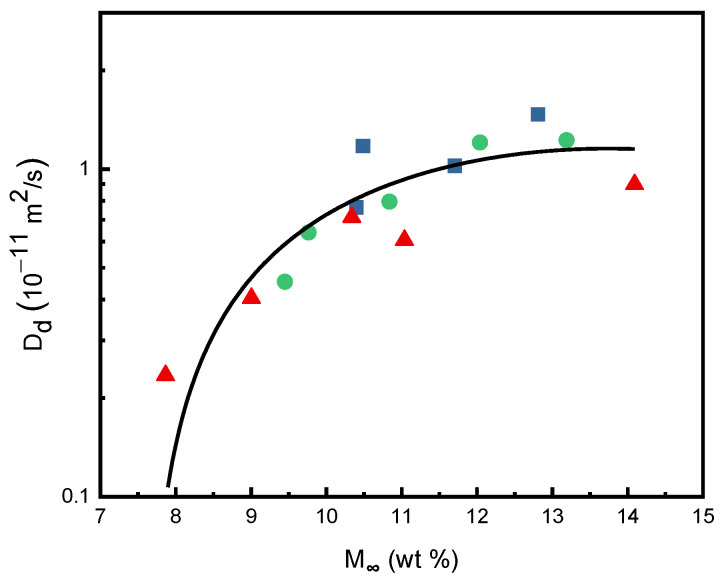
Plot of desorption diffusivity (*D_d_*) versus SCCO_2_ equilibrium mass gain fraction at various temperatures: ■, 40 °C; ●, 50 °C; ▲, 60 °C.

**Figure 6 molecules-29-04233-f006:**
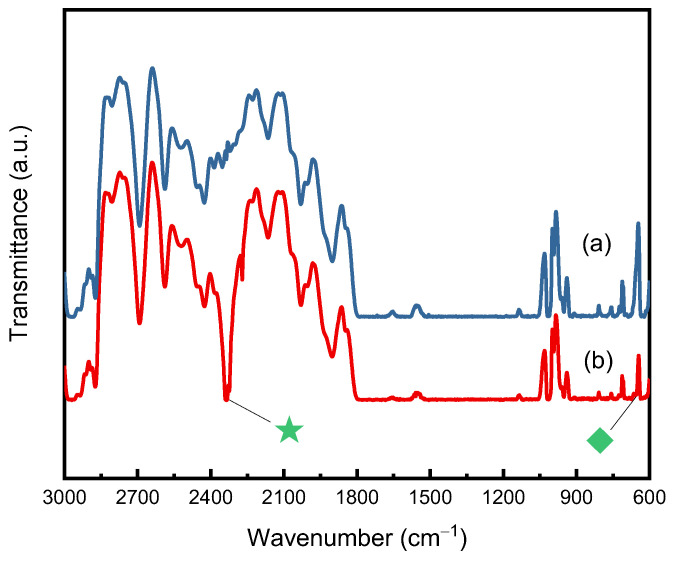
FTIR spectra for the antisymmetric stretching mode (*ν*_3_) for: (**a**) untreated PEI; and (**b**) the SCCO_2_-treated PEI after 24 h of desorption. (The sorption process was operated at 20 MPa, 40 °C for 12 h. The green symbols represent wavenumbers at 2340 cm^−1^ and 660 cm^−1^, respectively.).

**Figure 7 molecules-29-04233-f007:**
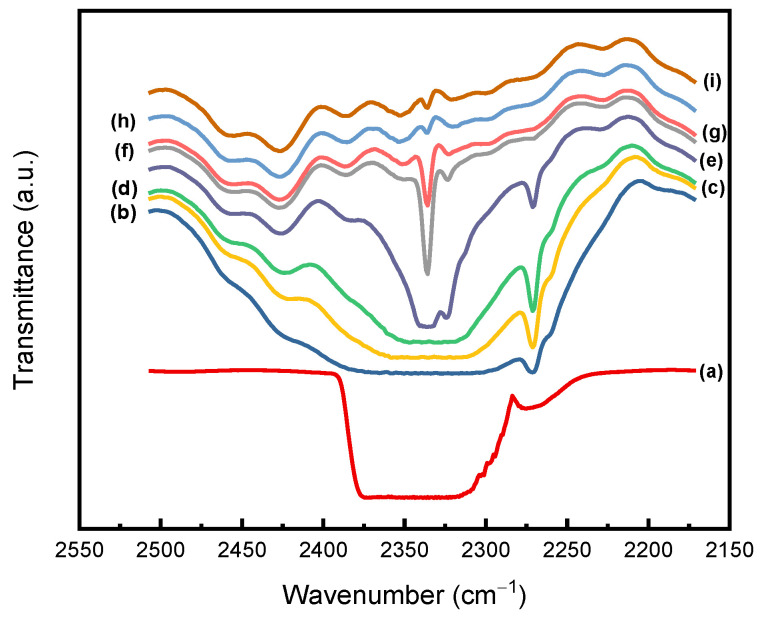
FTIR spectra for the antisymmetric stretching mode (*ν*_3_) of CO_2_ for (**a**) gaseous CO_2_; (**i**) untreated PEI; and the CO_2_ entrapped within PEI film after various desorption times of (**b**) 120 s; (**c**) 1 h; (**d**) 4 h; (**e**) 24 h; (**f**) 48 h; (**g**) 72 h; and (**h**) 96 h. (The sorption process was operated at 20 MPa, 40 °C for 12 h).

**Figure 8 molecules-29-04233-f008:**
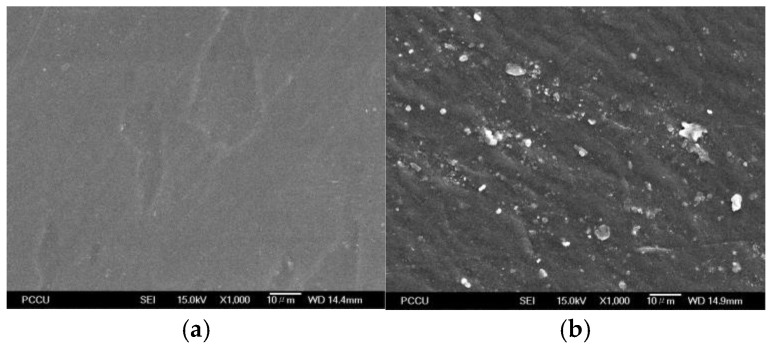
The SEM images of (**a**) untreated PEI and (**b**) SCCO_2_-treated PEI at 40 °C, 20 MPa, and sorption time of 12 h.

**Figure 9 molecules-29-04233-f009:**
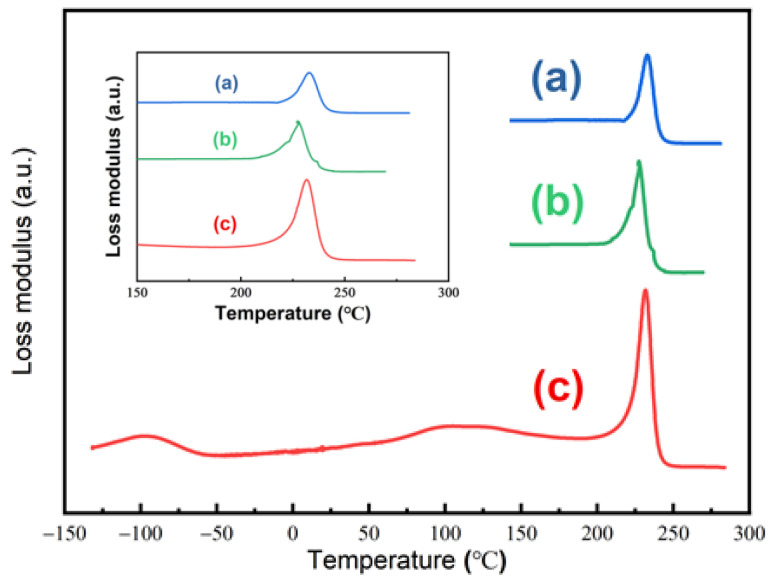
Loss modulus as a function of the temperature of PEI for: (**a**) untreated; (**b**) treated with SCCO_2_ at 20 MPa and 40 °C for 12 h; (**c**) more than one month of desorption with the same treated conditions as (**b**).

**Figure 10 molecules-29-04233-f010:**
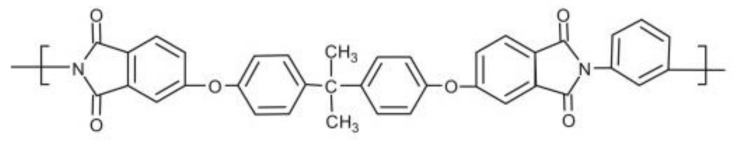
The structure of polyetherimide (PEI).

**Figure 11 molecules-29-04233-f011:**
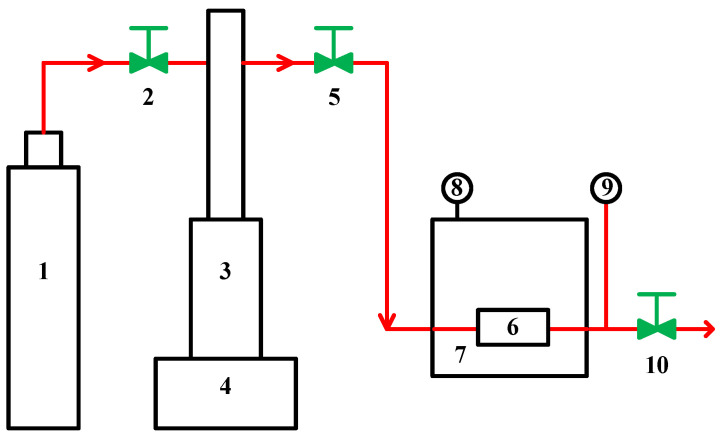
Schematic diagram of the experiment apparatus. 1. CO_2_ gas cylinder; 2. check valve; 3. high pressure syringe pump; 4. temperature and pressure controller; 5. needle valve; 6. high pressure equilibrium cell; 7. constant temperature bath; 8. temperature indicator; 9. pressure indicator; 10. needle valve.

**Figure 12 molecules-29-04233-f012:**
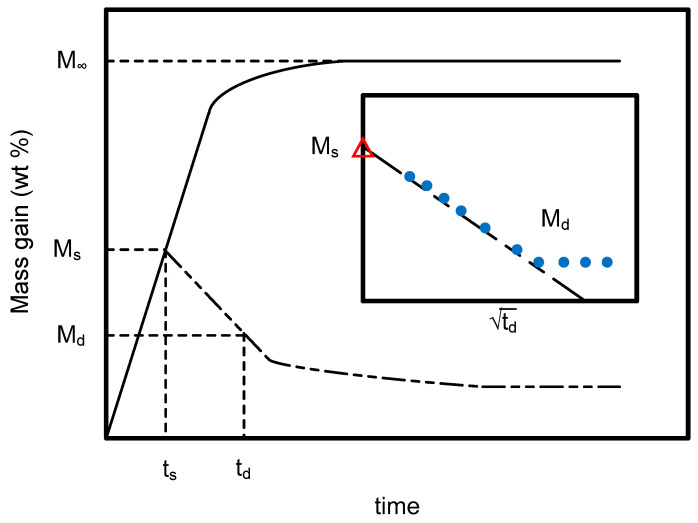
Schematic illustration for the sorption (―) and desorption (― ‒ ‒ ―) profiles. The inserted graph shows that the mass gain fraction decreased linearly with the square root of the initial desorption time.

**Table 1 molecules-29-04233-t001:** Experimental results of SCCO_2_ density, equilibrium sorption mass gain fraction (*M_∞_*), desorption diffusivity (*D_d_*), and sorption diffusivity (*D_s_*) at various temperatures and pressures.

Pressure (MPa)	Temperature (°C)	SCCO_2_ Density (kg/m^3^)	*M_∞_* (wt%)	*D_d_* (10^−11^m^2^/s)	*D_s_* (10^−11^m^2^/s)
20	40	839.8	10.5	1.18	0.13
20	50	784.3	9.8	0.64	0.18
20	60	723.7	9.0	0.40	0.30
30	40	909.9	11.7	1.02	0.11
30	50	870.4	10.8	0.79	0.24
30	60	829.7	10.3	0.71	0.22
40	40	956.1	12.8	1.47	0.12
40	50	923.3	12.0	1.20	0.12
40	60	890.1	11.0	0.61	0.26

## Data Availability

The data are contained within the article.

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
