# Peer review of "Experimental Study for the Sorption and Diffusion of Supercritical Carbon Dioxide into Polyetherimide"

_molecules, 2024, doi:10.3390/molecules29174233_

Round 1

Reviewer 1 Report

Comments and Suggestions for Authors

Comments from Reviewer

Manuscript ID: molecules-molecules-3158000-peer-review-v1

Experimental Study for the Sorption and Diffusion of Supercritical Carbon Dioxide into Polyetherimide

The authors discussed the sorption and diffusion of SCCO2 into PEI membrane samples in the temperature and pressure range from 40 °C and 20 MPa to 60 °C and 40 MPa.

The current form's presentation of methods and scientific results is satisfactory for publication in the Molecules journal. Some comments apply to the entire article. Please take this into account when making corrections. The minor and significant drawbacks to be addressed can be specified as follows:
Minor comments:
1.    Fig. 1, data for 52h. the symbol “+” in invisible.
2.    Line 115, Ms. In Fig. 4 the data for Minfinty are collected.
3.    Figs. 1 and 2, y-axis, title. “Wt” or “wt”?
4.    Fig. 6. Why were green symbols introduced? Please explain the figure description.
5.    Fig. 12., lines and figure captions. The application of the similar dashed lines for desorption profiles and for showing Ms and Md is confusing.
6.    Line 301. 3.2. Apparatus and Procedures ---> 3.2. Apparatus and procedures
7.    Fig. 11. The same font size for "1", "2", ... should be used.
Major comments:
1.    Lines 115-117. Figures in the text should be listed sequentially. Mentioning Figs. 4 and 12 immediately after Fig. 1 are confusing.
2.    Fig. 3. Why is the data not described with a straight line for 40oC? The same comment for 50oC.
3.    Fig. 4. Introducing trend lines can sometimes lead to strange suggestions. Do the authors suggest that for lower densities (40oC) Minfinity will grow?
4.     did the authors study adsorption/desorption cycles? Polymeric materials tend to swell and become permanently deformed as a result of the adsorption process.
Sincerely,
    The reviewer.

Reviewer 2 Report

Comments and Suggestions for Authors

The manuscript reports an experimental investigation of the sorption and diffusion of super-critical carbon dioxide in Polyetherimide at 40, 50 and 60 °C and three pressures 30, 40 and 50 MPa. The authors resorted to the same experimental setup and protocol proposed in past contributions (Journal of Applied Polymer Science, Vol. 94, 474–482 (2004); J. of Supercritical Fluids 28 (2004) 207–218). Specifically, they used a gravimetric offline approach to measure the variation of the specimen mass during sorption and desorption tests. The work is interesting since solubility and diffusivity measurements of supercritical carbon dioxide in polymers are of interest in several applications and these data are missing in the literature for polyetherimide.

The following comments should be addressed by the authors before the manuscript could be accepted for publication.

Abstract

1-      It should be clearly stated that the ‘gravimetric method’ is used ex situ. Generally speaking, the gravimetric method refers to the measurement of the sample mass in situ, i.e. during sorption or desorption (see for instance the magnetic suspension balance used in Journal of Supercritical Fluids 16 (1999) 81–92).

2-      The crossover phenomenon is reported as an important result of the present study. However, in figure 4, this phenomenon is not evident. Instead, it appears that when the supercritical fluid density exceeds 893 kg/m3 the three solubility curves are superimposed.

3-      The IR analysis of the specimen during desorption of carbon dioxide is qualitative and does not add any relevant information regarding the solubility and diffusivity of carbon dioxide in PEI.

Results

1- At fixed temperature and pressure, it appears that in Figure 1 the sorption/desorption cycles were conducted consecutively in terms of sorption time on the same specimen. The authors should confirm this aspect. The sorption kinetics, and consequently the sorption diffusivities, evaluated from the extrapolated Ms values could be severely affected by the polymer swelling/deswelling produced during each sorption/desorption cycle. Therefore, the authors should describe in more detail the experimental protocol and specifically: how many samples were used to conduct the tests; the history of sorption/desorption cycles chosen for each sample.

2-   More accurate density values of supercritical carbon dioxide can be found in the NIST database (see https://webbook.nist.gov/chemistry/fluid/). Figure 4 should be replotted based on these values ant table 1 should be modified accordingly.

3-   In reference 44, figure 6, the crossover is clearly highlighted by a change of the curve concavity when the solubility data are plotted as a function of pressure or density of supercritical carbon dioxide. The crossover is absent in figure 3 of the present work. In figure 4, the comment 2 (abstract) above applies. In addition, there is no evident change of the curve concavity. In my opinion, the polymer undergoes plasticization as stated by the authors but its proof should be recognized in the slope change of the solubility curves reported in Figure 4. The authors are referred to the work of Pierleoni et al. J. Phys. Chem. B 2017, 121, 42, 9969–9981 (https://doi.org/10.1021/acs.jpcb.7b08722) for the sake of comparison. What is the uncertainty in the solubility measurements?

4-  Lines 217-219, the strong interaction between carbon dioxide and the carbonyl groups should increase the solubility and decrease the penetrant diffusivity. Conversely, the authors stated that it would increase the penetrant diffusivity. The interaction of carbon dioxide with PEI is weak but is expected to be greater than with PC.  The authors are referred to the work of Scherillo et al. Macromolecules 2022, 55, 24, 10773–10787 (https://doi.org/10.1021/acs.macromol.2c01382) for the sake of comparison.

5- The IR analysis should be conducted on the polymer signals to highlight the occurrence of polymer plasticization. The IR analysis of carbon dioxide in PEI is only qualitative and does not add any relevant information on the polymer plasticization. Lines 259-261, “The phenomena in Figure 7 show that CO2 remained in PEI, which might be the reason for the plasticization of PEI” is not clear and seems to be irrelevant.

Materials and Methods

1-      Eq. 1 would result in negative values of the mass gain fraction as it is defined and the sign should be changed.

Minor Revisions

1-      In figure 2, the square root of time should be reported on the x-axis.

2-      In figure 4 and 5, it is not clear how the continuous lines were obtained.

3-   In figure 4 the data are plotted against the fluid density not the pressure

Based on the previous comments, I recommend major revisions for the manuscript.

Comments on the Quality of English Language

English is satisfactory but can be improved. Some typos were found in the manuscript. 

Reviewer 3 Report

Comments and Suggestions for Authors

The authors of the study “Experimental Study for the Sorption and Diffusion of Supercritical Carbon Dioxide into Polyetherimide” presented experimental results and analysis for the SCCO2-assisted polyetherimide (PEI) processing. They stablished relationship between the sorption and desorption processes of SCCO2 according to varying temperature and pressure. Major comments that could be considered by the authors before a possible publication are as follows:

The description and discussion of the  supercritical CO2 (SCCO2)-assisted polymer processing, causing surface modification of the material, would be more complete if compared to other techniques employed for the same goal. Could the authors add the comparison to other techniques in the introduction?

The manuscript structure is non-conventional. In the beginning of the Results section, the text refers to equation 5, Figures 4 and 12 without presenting them. This makes the readability of the manuscript very poor. It is not intuitive to readers.

Authors discussed the numerical values obtained for 𝑀 before presenting the parameter and without discussing its physical meaning.

What is the meaning of the crossover point obtained from the analysis of the equilibrium sorption amount of CO2 versus pressure at different temperatures? What would mean a higher or a lower crossover point?

Why does the diffusion of CO2 during sorption and desorption process are lower for stronger bonding forces between coiled polymer chains? What is more important for the diffusion of CO2, the structure of the polymer chain (coiled, tangled, etc.) or the force of the bonding between then?

Authors could discuss more the SEM images. Also, why the morphology changes clear by the SEM imagens indicate the  plasticization effect on the polymer?

Round 2

Reviewer 1 Report

Comments and Suggestions for Authors

The authors have made a substantial improvement for this article. The manuscript can be accepted for publishment in the present form.

Reviewer 2 Report

Comments and Suggestions for Authors

The Auhtors replied to the Reviewer's comments appropriately and modified the manuscript accordingly. The manuscript can now be accepted for publication.

Reviewer 3 Report

Comments and Suggestions for Authors

The manuscript can be published in the present form.